## [Decision Letter · Decision Letter 0]

19 Feb 2025

PONE-D-24-57315Helpful assistant or fruitful facilitator? Investigating how personas affect language model behaviorPLOS ONE

Dear Dr. Luz de Araujo,

Thank you for submitting your manuscript to PLOS ONE. After careful consideration, we feel that it has merit but does not fully meet PLOS ONE’s publication criteria as it currently stands. Therefore, we invite you to submit a revised version of the manuscript that addresses the points raised during the review process.

We look forward to receiving your revised manuscript.

Kind regards,

Jan Christopher Cwik, Ph.D.

Academic Editor

PLOS ONE

2. Thank you for stating the following financial disclosure:  [This research has been funded by the Vienna Science and Technology Fund (WWTF) [10.47379/VRG19008] “Knowledge-infused Deep Learning for Natural Language Processing”.].  Please state what role the funders took in the study.  If the funders had no role, please state: "The funders had no role in study design, data collection and analysis, decision to publish, or preparation of the manuscript." If this statement is not correct you must amend it as needed.

Additional Editor Comments:

I want to thank all reviewers for their time and effort in reading this manuscript. As you can see from the review below, all reviewers pointed out several mainly methodological aspects that must be considered in revising the manuscript.

Reviewers' comments:

Reviewer's Responses to Questions

**Comments to the Author**

1. Is the manuscript technically sound, and do the data support the conclusions?

Reviewer #1: Yes

Reviewer #2: Partly

Reviewer #3: Yes

2. Has the statistical analysis been performed appropriately and rigorously? 

Reviewer #1: Yes

Reviewer #2: Yes

Reviewer #3: Yes

3. Have the authors made all data underlying the findings in their manuscript fully available?

Reviewer #1: Yes

Reviewer #2: Yes

Reviewer #3: Yes

4. Is the manuscript presented in an intelligible fashion and written in standard English?

Reviewer #1: Yes

Reviewer #2: Yes

Reviewer #3: Yes

5. Review Comments to the Author

Reviewer #1: Thanks for the opportunity to read and comment on this paper. I think this is an interesting and certainly relevant study that could be published after some revisions have been made.

Major comments

1) Selection of LLMs: The authors use seven different large language models in their analysis – seven out of a much larger collection of potentially relevant LLMs contained in the Transformers library. Unfortunately, the manuscript does not provide a clear explanation of justification for the selection of just these models. Why not one of the Llama models, for example? Case selection – in this case, LLMs are different cases to be studied – can have massive effects on a study’s results, so the authors choices here really need to be made transparent and justified. Readers usually want a very convincing assurance that the selection of just these models does not skew the results in any way.

2) Human coders: To investigate their RQ3, the authors compare simulated LLM attitudes and perceptions of racism and toxicity of a set of tweets to corresponding data from humans. The problem here is that the authors only vaguely refer to a previous study from where the human data were (seemingly) taken (?), which this is not enough. The origin and attributes of the human attitude & tweet evaluation data need to be explained in more detail so that readers can understand what the authors are working with, what the potential weaknesses or shortcomings of these data are, and how the choice for those data could potentially influence the authors’ results (are there alternatives?). All this should be transparent even when one has not read the other study in question.

Minor comments

1) The graphs appear very grainy on my end and are difficult to read. Please upload them in a higher resolution and/or different file format.

2) Some of the real-life personas that are used in the analyses (e.g., Rosa Luxemburg, Jörg Haider) are not introduced. I would not assume that everyone knows who Rosa Luxemburg was, and Jörg Haider is likely an even bigger unknown to many outside of (central/German-speaking) Europe. I would introduce them at least briefly in footnotes.

3) On page 2, I would briefly define the term “toxicity” – it does become clear what this means from the context, but I still briefly stumbled over the term when reading.

Reviewer #2: • Line 6 “burst the research” is a non-scientific expression. Rephrase it.

• Line 78 – 90. This conclusion appears soon after the rational of the study, which reflect a jump for the reader. It appears not at a right place. Also recheck the claim language, ‘first study to comprehensively investigate’ the impact of personas on LLM behaviour”. There are few published studies, please check if scope of the study similar or different.

• Line 94 to 99. Please add more references.

• Line 254 – 257 how did you infer that statement? Can you show statistics in parenthesis?

• Can you please indicate where Persona Configuration or variable details are mentioned in this article?

• Can you please indicate what was the criteria for Personas programming and testing?

• Did you use any Scale of measuring ‘the user satisfaction’ If yes can you indicate, if NO can you indicate why not?

Reviewer #3: Dear authors,

Your article titled “Helpful assistant or fruitful facilitator? Investigating how personas affect language model behavior” addresses an important and timely topic, exploring the impact of assigned personas on the behavior of large language models (LLMs). The study is well-structured, clearly written, and provides valuable insights into the ways personas influence LLM outputs across various dimensions, including task performance, biases, social attitudes, and refusal rates.

Summary of the paper

Your paper investigates how assigning different personas to LLMs affects their responses across multiple dimensions. Specifically, you assign 162 personas from 12 categories (e.g., gender, sexual orientation, occupation) to seven different LLMs and prompt them with questions from five datasets. These datasets cover both objective (e.g., history, math) and subjective tasks (e.g., beliefs, values). The paper introduces a control persona setting (paraphrases of "helpful assistant") and an empty persona setting to isolate the effects of assigned personas.

The findings demonstrate that persona assignments lead to significant variability in responses across models, with some patterns of persona behavior generalizing across LLMs. You provide a rigorous quantitative analysis of task performance, social biases, and model refusal rates. The work contributes meaningfully to the literature by highlighting the inconsistencies in persona effects, particularly in how LLMs respond differently to personas with similar demographic traits.

Suggested revisions

1. Clarification on persona effects: While the results effectively show that different personas yield varying performances across models, an expanded discussion on why these differences occur would strengthen the argument. How could architectural differences between LLMs (e.g., ChatGPT vs. Mixtral) contribute to these disparities? What implications might these differences have for future model updates? This discussion would help clarify whether the differences stem from training data, fine-tuning methodologies, or other factors.

2. Contextualization of refusals: The paper provides compelling evidence of disparities in refusal rates across personas. However, a deeper exploration of the ethical and practical implications of these refusals would be useful. How do these refusal patterns impact fairness, safety, and trust in LLMs? Discussing these concerns would provide a stronger contextual grounding for your findings.

3. Recommendations for stakeholders: Given the study’s findings, it would be beneficial to provide explicit (short) recommendations for different stakeholders:

o Model developers: How should developers address inconsistencies in persona performance and biases?

o Policy and safety regulators: What considerations should be made in regulating persona-based interactions in LLMs?

o Users: How can users be made aware of potential biases in persona-based LLM interactions?

4. Minor suggestions for clarity and rigor:

a) Clarification on Sample Size: It is unclear to me how many times each question was submitted per persona-model combination. Given that LLMs generate probabilistic outputs, could running multiple iterations (e.g., 50 per persona-model pairing) and using modal responses improve robustness? If this was done, please clarify in the text; if not, discuss whether it could be a useful methodological refinement for future research.

b) Impact of system messages: Section 3 mentions some models receive system messages while others do not. Could this distinction introduce biases? Would the same models produce different results if all were tested with or without system messages?

c) Control persona paraphrases: Table 2’s control personas include some paraphrases (e.g., “competent second-in-command”) that may not be semantically equivalent to “helpful assistant”. Could this introduce unintended variation in results? Discuss whether this might contribute to arbitrary or disparate findings that are discussed later in the paper.

d) Interpretation of Table 3: The sentence on lines 254–255 states “Table 3 shows...” but does not explain why the results appear as they do. Also, Table 3 suggests that, overall, technology personas outperform others. Are these differences statistically significant? Providing an explanation might strengthen the interpretation.

e) Clarification in Tables 5, 6, 7: In row 490 and other references, should “homosexual” be replaced with “heterosexual” to align with the categories listed? “Gay” is already a category, but “heterosexual” does not seem to be listed.

f) Statistical significance in Figures 6 and 9: Pearson correlation values are presented, but it is unclear which are statistically significant. If feasible and you feel this would add value to the results, consider highlighting coefficients that meet the 1% significance level with asterisks to improve interpretability (but this is a minor/optional suggestion).

6. PLOS authors have the option to publish the peer review history of their article (what does this mean?). If published, this will include your full peer review and any attached files.

Reviewer #1: No

Reviewer #2: **Yes: **Kanwal Qayyum

Reviewer #3: No

---

## [Author Response · Author response to Decision Letter 1]

7 Mar 2025

We thank the reviewers for the constructive and insightful comments. We have carefully considered all the feedback and suggestions. Our detailed responses to each point are provided in the cover letter, and all corresponding changes are highlighted in the 'Revised Manuscript with Track Changes' file. We also provide in a separate document ("response to reviewers") a copy of the responses in the cover letter.

---

## [Decision Letter · Decision Letter 1]

19 May 2025

Helpful assistant or fruitful facilitator? Investigating how personas affect language model behavior

PONE-D-24-57315R1

Dear Dr. Luz de Araujo,

We’re pleased to inform you that your manuscript has been judged scientifically suitable for publication and will be formally accepted for publication once it meets all outstanding technical requirements.

Kind regards,

Jan Christopher Cwik, Prof. Dr. Dr.

Academic Editor

PLOS ONE

Reviewers' comments:

Reviewer's Responses to Questions

**Comments to the Author**

1. If the authors have adequately addressed your comments raised in a previous round of review and you feel that this manuscript is now acceptable for publication, you may indicate that here to bypass the “Comments to the Author” section, enter your conflict of interest statement in the “Confidential to Editor” section, and submit your "Accept" recommendation.

Reviewer #1: All comments have been addressed

Reviewer #2: All comments have been addressed

Reviewer #3: All comments have been addressed

2. Is the manuscript technically sound, and do the data support the conclusions?

Reviewer #1: Yes

Reviewer #2: Yes

Reviewer #3: Yes

3. Has the statistical analysis been performed appropriately and rigorously? 

Reviewer #1: Yes

Reviewer #2: Yes

Reviewer #3: Yes

4. Have the authors made all data underlying the findings in their manuscript fully available?

Reviewer #1: (No Response)

Reviewer #2: Yes

Reviewer #3: Yes

5. Is the manuscript presented in an intelligible fashion and written in standard English?

Reviewer #1: Yes

Reviewer #2: Yes

Reviewer #3: Yes

6. Review Comments to the Author

Reviewer #1: Many thanks for the opportunity to read and comment on this manuscript again! My earlier comments are addressed in principle (even though the selection of LLMs and the survey data could still be explained in more detail).

Reviewer #2: All comments are appropriately addressed. This article will be a good addition in scientific knowledge.

Reviewer #3: I thank the authors for their detailed responses and for clearly highlighting the changes made. I have reviewed each point in the response letter and verified the corresponding revisions in the manuscript. My initial suggestions have been addressed appropriately. The authors have made thoughtful revisions that improve the manuscript's clarity and value to the field. I am satisfied with the changes.

7. PLOS authors have the option to publish the peer review history of their article (what does this mean?). If published, this will include your full peer review and any attached files.

Reviewer #1: No

Reviewer #2: **Yes: **Kanwal Qayyum

Reviewer #3: No

---

## [Editor Report · Acceptance letter]

PONE-D-24-57315R1

PLOS ONE

Dear Dr. Luz de Araujo,

I'm pleased to inform you that your manuscript has been deemed suitable for publication in PLOS ONE. Congratulations! Your manuscript is now being handed over to our production team.

Kind regards,

on behalf of

Prof. Dr. Dr. Jan Christopher Cwik

Academic Editor

PLOS ONE